

# Construction and validation of a nomogram model to predict the poor prognosis in patients with pulmonary cryptococcosis

Xiaoli Tan[1,*], Yingqing Zhang[1,*], Jianying Zhou[2], Wenyu Chen[1] and Hua Zhou[2]

[1] Department of Respiratory, The Affiliated Hospital of Jiaxing University, Jiaxing, China
[2] Department of Respiratory, The First Affiliated Hospital, Zhejiang University School of Medicine, Hangzhou, China
[*] These authors contributed equally to this work.

## ABSTRACT

**Background**. Patients with poor prognosis of pulmonary cryptococcosis (PC) are prone to other complications such as meningeal infection, recurrence or even death. Therefore, this study aims to analyze the influencing factors in the poor prognosis of patients with PC, so as to build a predictive nomograph model of poor prognosis of PC, and verify the predictive performance of the model.

**Methods**. This retrospective study included 410 patients (78.1%) with improved prognosis of PC and 115 patients (21.9%) with poor prognosis of PC. The 525 patients with PC were randomly divided into the training set and validation set according to the ratio of 7:3. The Least Absolute Shrinkage and Selection Operator (LASSO) algorithm was used to screen the demographic information, including clinical characteristics, laboratory test indicators, comorbidity and treatment methods of patients, and other independent factors that affect the prognosis of PC. These factors were included in the multivariable logistic regression model to build a predictive nomograph. The receiver operating characteristic curve (ROC), calibration curve and decision curve analysis (DCA) were used to verify the accuracy and application value of the model.

**Results**. It was finally confirmed that psychological symptoms, cytotoxic drugs, white blood cell count, hematocrit, platelet count, CRP, PCT, albumin, and CD4/CD8 were independent predictors of poor prognosis of PC patients. The area under the curve (AUC) of the predictive model for poor prognosis in the training set and validation set were 0.851 (95% CI: 0.818-0.881) and 0.949, respectively. At the same time, calibration curve and DCA results confirmed the excellent performance of the nomogram in predicting poor prognosis of PC.

**Conclusion**. The nomograph model for predicting the poor prognosis of PC constructed in this study has good prediction ability, which is helpful for improving the prognosis of PC and further optimizing the clinical management strategy.

Corresponding authors
Wenyu Chen, 00135116@zjxu.edu.cn
Hua Zhou, zhouhua1@zju.edu.cn

## INTRODUCTION

Pulmonary cryptococcosis (PC) is a pulmonary fungal disease caused by cryptococcal infection. It usually occurs in people with immune insufficiency, but in recent years, its incidence rate also shows a gradually rising trend in people with normal immune function (*Fisher, Valencia-Rey & Davis, 2016*; *Rigby & Glanville, 2012*; *Willenburg & Hadley, 2009*). The central nervous system infection caused by cryptococcus accounts for 80% of cryptococcal infection (*Sheng & Liu, 2019*). The most serious clinical manifestation of PC is cryptococcal meningitis, with a mortality rate of 20% to 60% (*Molloy et al., 2017*; *Zhang, Tan & Tian, 2020*). In addition, most patients with cryptococcal infection are also accompanied by some underlying diseases, such as hematology malignance, HIV infection, and other immunodeficiency diseases. However, at present, most comorbidity studies focused on cryptococcal meningitis in immunocompromised populations (*Setianingrum, Rautemaa-Richardson & Denning, 2019*; *Tsai et al., 2018*; *Zhang, Tan & Tian, 2020*). There are individual differences in clinical manifestations of PC, and the clinical characteristics and prognosis of patients with PC of different serotypes and genotypes are also significantly different (*Ponzio et al., 2019*; *Tsai et al., 2019*). Different clinical manifestations and treatment methods directly affects the treatment effect of PC (*Sloan & Parris, 2014*; *Williamson et al., 2017*). Therefore, the treatment strategies of PC should be tailed based on patients' conditions. Furthermore, the non-specific manifestations of PC may make timely diagnosis challenging. If patients are not actively treated, it is more likely to lead to disseminated infection or other complications, even death and other poor prognosis results. However, there is a lack of research on adverse prognostic factors of PC, especially the lack of models for predicting poor prognosis.

The nomogram is a predictive tool widely used in medicine, which generates individual probabilities in clinical events by integrating different prognosis and decisive variables to meet our needs for biological or clinical models, and has great application value in personalized medicine (*Balachandran et al., 2015*). Compared with traditional prediction models, nomograms are easy to understand, fast to calculate, and more accurate, thus helpful for rapid and effective decision-making in clinical practice (*Ross et al., 2002*). In recent years, an increasing number of studies have used nomograms to predict the occurrence of clinical events or the prediction of disease risk, such as the estimation of long-term survival rate of patients with non-small cell lung cancer after surgery (*Wang et al., 2021*), the evaluation of renal anemia in patients with IgA nephropathy (*Li et al., 2021*) and the estimation of the risk of severe hand-foot-and-mouth disease in children (*Wang et al., 2019*) etc. Some studies have also adopted nomograms to assess the risk factors of disease prognosis (*Flanigan, Polcari & Hugen, 2011*; *Kattan, 2003*; *Touijer & Scardino, 2009*; *Yang, 2013*), but there are few reports in the prognostic research of cryptococcosis.

Through a retrospective study of the clinical data of patients with PC, we aim to screen out the factors influencing the prognosis of PC and establish a scoring system for poor prognosis based on the nomogram model, so as to improve the early diagnosis and treatment as well as the prognosis of PC.

## MATERIAL AND METHODS

### Ethics statement

Human studies were reviewed and approved by the Ethics Committee of the First Affiliated Hospital of Zhejiang University in China (ID: IIT20210761A), and the Institutional Review Board waived written informed consent. All procedures were conducted in accordance with the Declaration of Helsinki (1964).

### Data collection

A total of 525 patients diagnosed with PC in the First Affiliated Hospital of Zhejiang University from January 2008 to December 2019 were included in this study. General information of patients on admission was collected as follows: demographic information (gender, age, the number of days in hospital, BMI, onset time, state of diagnosis), information on clinical features (other lung effects, fever, header, twitch, nausea, cough, shortness of break, mental symptoms, asymptomatic, imaging), laboratory test indicators (white blood cell count, neutrophils, lymphocytes, hemoglobin, packed cell volume, platelet count, CRP, PCT, albumin, globulin, alanine aminotransferase, aspartate aminotransferase, creatinine, total number of lymphocytes, T cell, auxiliary T, lethality T, CD4/CD8, NK cell, B cell) comorbidities (diabetes, tuberculosis, malignant tumor, hematological malignancies, organ transport, other complications, HIV) and treatment methods (capsule antigen, hormone therapy, immunosuppressant, cytotoxic drugs). The inclusion criteria were as follows: (1) PC was diagnosed by pathology or histology; (2) the diagnosis year was from January 2008 to December 2019. Meanwhile, the exclusion criteria were as follows: (1) incomplete diagnostic data information; (2) hospitalization days<1 day; (3) discharge or death occurs when relevant examinations were not completed. In addition, the outcome indicators of this study were to evaluate whether the prognosis of PC patients was improved or not according to clinical symptoms and detection indicators. Significantly improved clinical symptoms and negative detection of the cryptococcal capsule antibody were considered as an improvement. No significant improvement in clinical symptoms and signs, other complications such as meningeal infection, or death of the patient were regarded as poor prognosis.

### Diagnostic criteria

Diagnostic criteria for cryptococcosis: Diagnostic methods for pulmonary cryptococcosis include histological examination, fungal culture, serum cryptococcal antigen examination, and imaging examination. Histological examination: when typical yeast cells with capsules, narrow necks, budding but no hyphae are observed in granulomas or jelly-like lesions in lung tissue, cryptococcosis is diagnosed. Microbiological examination: Capsulated yeasts are observed in sputum, bronchoalveolar lavage fluid, or tissue specimens, which indicates pulmonary cryptococcosis infection. Diagnosis can be confirmed by culturing cryptococci from sputum or other specimens. Serum cryptococcal antigen examination: Cryptococcal milk agglutination test can detect cryptococcal capsular antigen in cerebrospinal fluid, blood, pleural effusion, BALF and other specimens. Among people with impaired immune function, 56% to 70% are positive. However, for people with normal immune function,

antigen detection has poor sensitivity in diagnosing pulmonary infections. In addition, metagenomic next-generation sequencing (mNGS) can detect very low-abundance pathogenic bacteria and can be used for early detection of cryptococcal infections.

Diagnostic criteria for cryptococcal meningitis: A positive result in any of cerebrospinal fluid fungal smear, culture, and cryptococcal latex agglutination test can confirm the diagnosis of cryptococcal infection in the central nervous system. The patient's clinical symptoms, signs, and routine, biochemical, and imaging examinations of cerebrospinal fluid are of great significance for the diagnosis of cryptococcal meningitis.

## Statistical analysis

The whole dataset was divided into two groups: training set (70%, $n = 368$) and validation set (30%, $n = 157$). The training set was used to build the nomogram prediction model, and the validation set was used to verify the accuracy of the nomogram. Statistical analysis was performed using SPSS V23.0 (IBM, Armonk, NY, USA) and R software (*R Core Team, 2019*). The normal distribution of continuous variables was determined by the Kolmogorov–Smirnov test or Shapiro–Wilk test in SPSS. Continuous data was expressed as mean ± standard deviation or median (quartile distance), while categorical data was expressed as numbers or percentages. In univariate analysis, Student's *T*-test or Mann–Whitney U test was used for continuous variables, and Chi-square test or Fisher's exact test was employed for categorical variables. The least absolute shrinkage and selection operator (LASSO) method was used to screen the relevant variables that affect the prognosis of patients (*Lee et al., 2014*). According to the results of LASSO regression analysis, the screened independent predictors were introduced into the multivariate logistic regression analysis to build a predictive model nomogram. Finally, the ROC curve and DCA curve were used to verify the accuracy and utility of the prediction model. In all analyses, *p*-values less than 0.05 were considered statistically significant.

# RESULTS

## Baseline characteristics

A total of 525 patients were included in this study as research objects. Patients were randomly assigned to the training set (70%, $n = 368$) and validation set (30%, $n = 157$). In the training set, 297 patients (80%) had improved prognosis, while 71 patients (20%) had poor prognosis. In the validation group, 113 patients (72%) had improved prognosis, while 44 patients (28%) had poor prognosis. Table 1 and Table S1 shows the demographic characteristics and clinical characteristics of patients in the training set and validation set. In the training set, the median age of patients was 49 years old, and the average hospitalization time was 21 ± 18 days; the average BMI was 22.96 ± 7.16; the average onset time was 44 ± 82 days. There was no significant difference in clinical characteristics between patients in the training set and validation set ($p > 0.05$).

## LASSO regression analysis

LASSO regression analysis (Fig. 1A) and cross validation (Fig. 1B) were conducted with the patient's prognosis (improved/poor) as the dependent variable, and the patient's

**Table 1  Comparison of measuring variables between all patients with or without poor prognosis.**

| Factor | Poor prognosis | Improved prognosis | T/Z | P |
|---|---|---|---|---|
| Age | 48.93 ± 14.78 | 47.92 ± 15.27 | 0.644 | 0.52 |
| The number of days in hospital | 19 (9.75∼28) | 15 (7∼24) | −2.3 | 0.021* |
| BMI | 22.51 (17.89∼25.66) | 22.86 (19.89∼25.03) | −0.275 | 0.783 |
| Onset time | 20 (10.75∼40) | 30 (13∼36) | −0.226 | 0.821 |
| White blood cell count | 6.6 (5.3∼8.2) | 6.6 (5.5∼8.2) | −0.138 | 0.89 |
| Neutrophils | 70.17 (63.25∼77.83) | 70.88 (62.7∼77.4) | −0.007 | 0.995 |
| Lymphocytes | 17.75 (10.95∼25.03) | 17.47 (9.3∼21.2) | −1.327 | 0.185 |
| Hemoglobin | 126 (110.82∼142) | 102 (4.83∼118) | −8.272 | <0.001* |
| Packed cell volume | 39.75 (35.18∼43.3) | 45.58 (39.2∼120) | −7.158 | <0.001* |
| Platelet count | 210.5 (159∼263.5) | 108 (41.8∼194.86) | −8.223 | <0.001* |
| CRP | 6.6 (2.15∼22.63) | 37.7 (14.1∼187) | −9.824 | <0.001* |
| PCT | 0.11 (0.04∼3.89) | 3.86 (0.38∼5.81) | −7.065 | <0.001* |
| Albumin | 40.05 (36.1∼44.3) | 36.25 (0.31∼38.53) | −8.36 | <0.001* |
| Globulin | 27.5 (23.4∼30.98) | 30.59 (28.1∼36.3) | −5.532 | <0.001* |
| Alanine aminotransferase | 20 (12∼31) | 13 (1.7∼28.16) | −5.173 | <0.001* |
| Aspartate aminotransferase | 18 (14∼25.21) | 20 (14∼26.76) | −1.743 | 0.081 |
| Creatinine | 72 (59.75∼80) | 69 (25∼74.62) | −4.291 | <0.001* |
| Total number of lymphocytes | 2223.78 (2161.05∼2302.34) | 2140.72 (74∼2225.93) | −7.504 | <0.001* |
| T cell | 68.22 (65.19∼71.8) | 68.31 (65.04∼72.76) | −0.152 | 0.879 |
| Auxiliary T | 25.65 (22.93∼28.04) | 25.66 (23.25∼28.44) | −0.25 | 0.803 |
| Lethality T | 40.03 (37.64∼42.96) | 39.96 (37.28∼42.42) | −0.394 | 0.693 |
| cd4/cd8 | 1.24 (0.49∼2.02) | 1.48 (0.57∼2.34) | −1.924 | 0.054 |
| NK cell | 15.42 (13.36∼17.61) | 16 (13.87∼18.19) | −1.514 | 0.13 |
| B cell | 14.3 (12.91∼15.82) | 14 (12.66∼15.5) | −0.986 | 0.324 |

**Notes.**
*$P < 0.05$

demographic information, clinical characteristics, laboratory test indicators, comorbidities, treatment methods and other influencing factors as the independent variables to determine the independent predictive factors for poor prognosis of patients. Finally, we selected the value with the minimum validation error ($\lambda = 0.0064$) for variable screening, and obtained 32 variables with significant differences: gender, age, the number of days in hospital, onset season, fever, twitch, cough, shortness of breath, mental symptoms, asymptomatic, diabetes, tuberculosis, malignant tumor, organ transplant, other comorbidities, capsular antigen, cytotoxic drugs, white blood cell count, lymphocytes, hemoglobin, packed cell volume, platelet count, CRP, PCT, albumin, globulin, alanine aminotransferase, aspartate aminotransferase, creatinine, auxiliary T, lethality T, cd4/cd8, constant, and they were included in the subsequent multifactor logistic regression analysis.

## Logistic regression analysis

For the 32 factors that may affect the prognosis of the above screening results, the logistic regression model was adopted for further multivariate analysis (Table 2). The results showed that nine factors, including neurological symptoms (OR 2.904, 95% CI [1.199–7.029]),
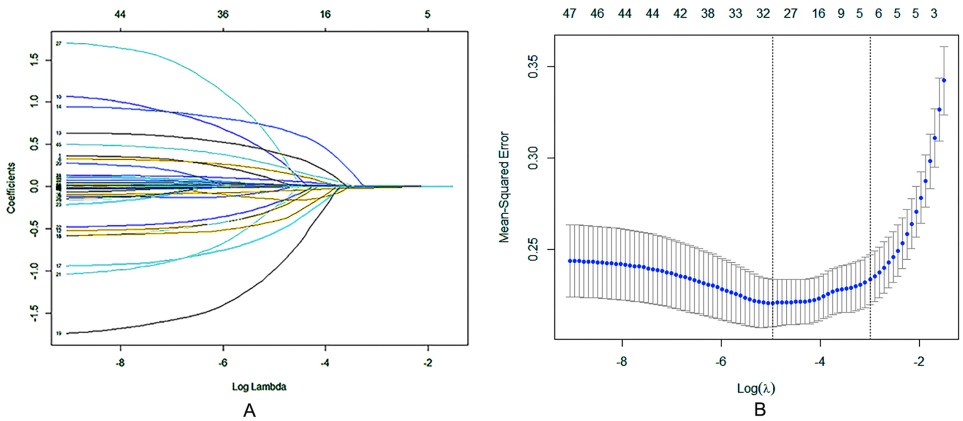

**Figure 1** **Regression analysis of influence factors based on Lasso for variable selection (A), and cross validation of the regression model (B).** Each curve in (A) represents the change trajectory of the coefficient of each independent variable, the coefficients of the unimportant variables tend to zero with the increasing of λ. For the cross validation, a confidence interval for the target parameter was obtained and marked by the two dotted lines, indicating two special ones of Lambda value.

hematocrit (OR 1.076, 95% CI [1.021–1.135]), white blood cell count (OR 1.148, 95% CI [1.026–1.285]), platelet count (OR 0.994, 95% CI [0.99–0.998]), CRP (OR 1.014, 95% CI [1.003–1.025]), PCT (OR 1.12, 95% CI [1.015–1.236]), albumin count (OR 0.918, 95% CI [0.865–0.974]), the ratio of CD4+ to CD8+ (OR 1.584, 95% CI [1.103–2.275]), and cytotoxic drugs (OR 5.342, 95% CI [1.195–23.881]), could significantly affect the poor prognosis of patients with PC.

Among them, the regression coefficient B of the two factors of albumin count and platelet count was <0, which were −0.086 and −0.006 respectively, indicating that the probability of poor prognosis would decrease with the increase of this index, while the occurrence or greater count of other indicators would increase the probability of poor prognosis.

## Construction and validation of nomogram prediction model

Nine independent influencing factors (psychological symptoms, hematocrit, white blood cell count, plate count, CRP, PCT, albumin count, the ratio of CD4+ to CD8+ and cytotoxic drugs) obtained by multifactor logistic regression analysis were used to construct a predictive nomograph of poor prognosis of PC patients (Fig. 2). The probability of poor prognosis can be estimated by calculating the total number of points from the vertical line of the variable to the scoring axis.

In this study, the ROC curve, calibration curve and DCA curve were used to verify the accuracy and utility of the prediction model. The area under the ROC curve (AUC) of the prediction model in the training set was 0.851 (95% CI: 0.818∼0.881), which was consistent with the C-index analysis results, indicating that the prediction model has good accuracy. The calibration curve showed that there was a good fit between the predicted risk and the actual risk of the model (Fig. 3B). Therefore, the risk prediction model has an ideal prediction effect in the training set.

**Table 2  Multivariate logistic regression analysis of risk factors.**

| Factors | | OR(95%CI) | P | Factors | | OR(95%CI) | P |
|---|---|---|---|---|---|---|---|
| Gender | Female | 1.459(0.746~2.851) | 0.269 | Organ transplant | Have | 0.371(0.044~3.103) | 0.36 |
| | Male | | 0 | | Without | Ref | 0 |
| Age | | 0.991(0.97~1.012) | 0.399 | Other comorbidities | Have | 0.629(0.305~1.298) | 0.21 |
| The number of days in hospital | | 0.991(0.975~1.008) | 0.305 | | Without | Ref | 0 |
| Onset season | | Ref | 0.061 | Capsular antigen | (+) | 1.034(0.529~2.021) | 0.922 |
| | Summer | 1.313(0.468~3.682) | 0.605 | | (-) | Ref | 0 |
| | Autumn | 3.02(1.218~7.486) | 0.017* | Cytotoxic drugs | Have | 5.342(1.195~23.881) | 0.028* |
| | Winter | 2.352(0.927~5.968) | 0.072 | | Without | Ref | 0 |
| | Spring | Ref | 0 | White blood cell count | | 1.148(1.026~1.285) | 0.016* |
| Fever | Have | 0.845(0.428~1.67) | 0.628 | Lymphocytes | | 0.979(0.948~1.012) | 0.214 |
| | Without | Ref | 0 | Hemoglobin | | 0.991(0.976~1.006) | 0.253 |
| Twitch | Have | 2.712(0.646~11.389) | 0.173 | Packed cell volume | | 1.076(1.021~1.135) | 0.006* |
| | Without | Ref | 0 | Platelet count | | 0.994(0.99~0.998) | 0.005* |
| Cough | Have | 0.597(0.31~1.148) | 0.122 | CRP | | 1.014(1.003~1.025) | 0.014* |
| | Without | Ref | 0 | PCT | | 1.12(1.015~1.236) | 0.024* |
| Shortness of breath | Have | 1.84(0.711~4.764) | 0.209 | Albumin | | 0.918(0.865~0.974) | 0.005* |
| | Without | Ref | 0 | Globulin | | 1.047(0.998~1.099) | 0.059 |
| Mental symptoms | Have | 2.904(1.199~7.029) | 0.018* | Alanine aminotransferase | | 0.984(0.966~1.003) | 0.099 |
| | Without | Ref | 0 | Aspartate aminotransferase | | 1.012(0.986~1.04) | 0.371 |
| Asymptomatic | No | 1.669(0.616~4.523) | 0.314 | Creatinine | | 0.996(0.989~1.003) | 0.278 |
| | Yes | Ref | 0 | Auxiliary T | | 0.986(0.943~1.03) | 0.525 |
| Diabetes | Have | 0.389(0.151~0.998) | 0.05* | Lethality T | | 1.029(0.985~1.075) | 0.2 |
| | Without | Ref | 0 | cd4/cd8 | | 1.584(1.103~2.275) | 0.013* |
| Tuberculosis | Have | 0.505(0.158~1.618) | 0.25 | Constant | | 0.054 | 0.239 |
| | Without | Ref | 0 | | | | |
| Malignant tumor | Have | 0.145(0.013~1.624) | 0.117 | | | | |
| | Without | Ref | 0 | | | | |

**Notes.**
*P < 0.05

To further investigate the clinical utility of the prediction model, a decision curve (DCA) (Fig. 3C) was drawn with the threshold probability of poor prognosis as the abscissa and the net benefit rate of patients as the ordinate. It can be clearly seen that patients whose prognosis was predicted based on the nomogram prediction scoring system have a higher net benefit than any patient predicted by a single index, indicating that the prediction model has clinical value for the training. The clinical impact curve (Fig. 3D) was drawn using the threshold probability of poor prognosis as the abscissa and the corresponding predicted number as the ordinate. The results showed that when the threshold probability $P < 0.6$, the number of people at high risk predicted by the model (red curve in the figure) was far more than the actual number of people with poor prognoses (blue curve). With the gradual increase of the threshold probability, the gap between the number of people predicted and the actual number of people with poor prognosis gradually narrowed, and

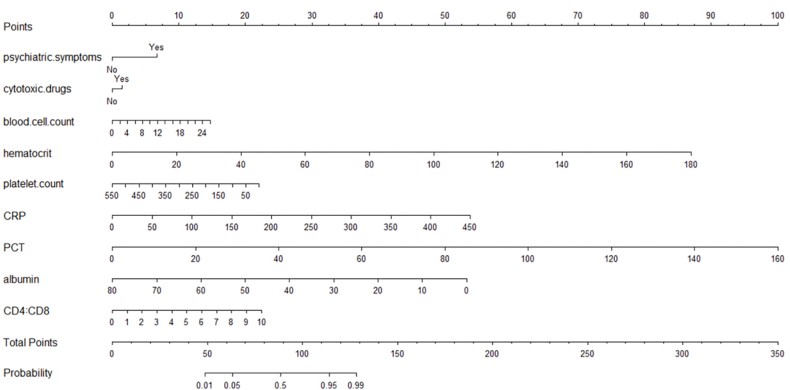

**Figure 2** The scores for all the parameters, and the probability of poor prognosis of PC corresponding to the total score.

when $P > 0.6$, the two were completely consistent. That is, when the probability of a poor prognosis predicted by the nomogram scoring model is greater than 60%, the prognosis of patients with PC can be accurately predicted.

# DISCUSSION

PC is a fungal infection of the lungs, which is closely related to the patient's immune function. *Yuchong et al. (2012)* analyzed 7,315 cases of cryptococci patients and found that only 17% of them had no underlying diseases, while most patients had underlying diseases such as HIV, liver disease, systemic lupus erythematosus or diabetes, which were mostly related to immune function. This is the reason why most of the research on cryptococcosis is based on AIDS patients or non-AIDS patients. In fact, cryptococcosis is becoming more and more likely to occur in people with normal immune function, and studies have shown that the proportion of patients with normal immune function in cryptococcosis patients in China is relatively high (*Zhang, Tan & Tian, 2020*; *Zhu et al., 2010*). In this study, 201 patients (38.3%) with cryptococcosis had other comorbidities, including 96 patients (18.3%) who were HIV positive.

The research subjects of this study are patients diagnosed with PC, whose main clinical manifestations include headache, fever, convulsions, nausea, cough, shortness of breath and neurological symptoms. Studies have shown that, except for neurological symptoms, there is no significant correlation between other symptoms and poor prognosis of PC, which is basically consistent with the findings of *Zhang, Tan & Tian (2020)* In this study, 55 patients (10.5%) had neurological symptoms, and 16 of them (29.1%) had a poor prognosis. Multivariate regression analysis showed that neurological symptoms ($p = 0.018$) were an independent factor influencing the poor prognosis of patients. The occurrence of neurological symptoms is an important sign of impaired brain function (*Esher, Zaragoza & Alspaugh, 2018*), which may be related to meningeal infection in patients with PC. In the case samples of this study, 317 cases (60.4%) were combined with meningeal infections, indicating that patients with PC complicated with cryptococcal meningitis are more likely

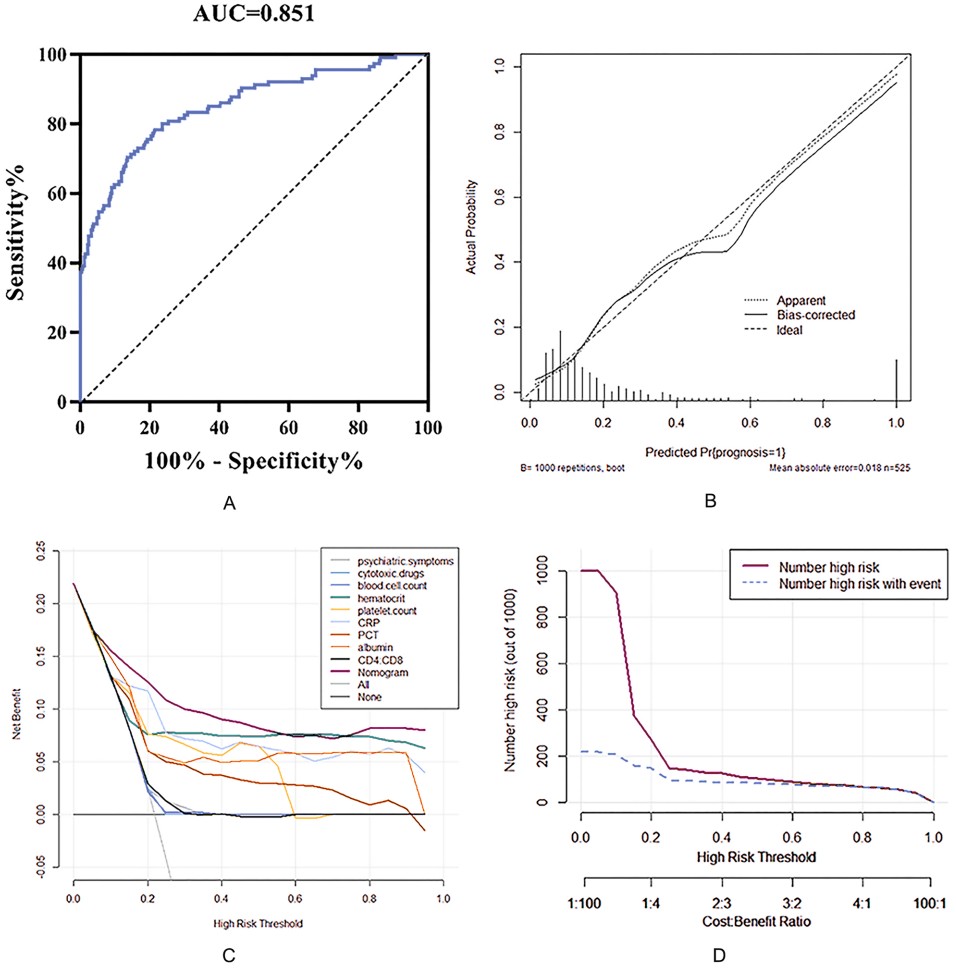

**Figure 3  Nomogram model to estimate the risk of meningeal infection in patients with pulmonary cryptococcosis.** (A) Receiver operating characteristic (ROC) curve of the nomogram, AUI, area under the ROC curve. (B) Calibration of the evaluation for nomogram model. (C) Comparison of the net benefit for patients with predictive model by the decision curves analysis (DCA). (D) Clinical value under different risk threshold probability.

to have a poor prognosis, and the occurrence of neurological symptoms can be used as an indicator of prognosis. It has been reported that pulmonary cryptococcal meningitis is the most serious condition of cryptococcosis, with a higher mortality rate and a higher probability of poor prognosis (*Pasquier et al., 2018*; *Williamson et al., 2017*; *Zhu et al., 2002*). Bahar et al. found that patients with cryptococcal meningitis may have a severely poor prognosis and risk of hearing impairment, muscle atrophy, and cognitive impairment (*Kashef Hamadani et al., 2018*).

Some studies have found that patients with PC who have been followed up for 2–11 years have a good prognosis in the HIV-negative population, with no death, recurrence or disseminated infection (*Yu et al., 2012*). However, acute severe dyspnea can affect the outcome of cryptococcosis treatment regardless of the presence or absence of immune

deficiency disease (*Vilchez et al., 2001*). A 10-year retrospective study found that 33% of PC patients without AIDS had symptoms of dyspnea, and the mortality rate was up to 55% (*Vilchez et al., 2001*). Nevertheless, 53 cases (10.1%) had shortness of breath symptoms in the samples of this study, but there was no significant difference in poor prognosis ($p = 0.209$), which may be related to the earlier onset of dyspnea symptoms. In the retrospective analysis of cases, dyspnea occurred in almost all cases within 24 h of admission (*Vilchez et al., 2001*).

The occurrence and prognosis of cryptococcosis are related to patients' underlying diseases or other causes, for example, diabetes will increase the risk of death from PC (*Archuleta et al., 2021*). The use of high concentration of glucocorticoids or immunosuppressants in patients with organ transplantation or malignant tumor will increase the incidence of cryptococcosis and the risk of poor prognosis (*Setianingrum, Rautemaa-Richardson & Denning, 2019*). Pulmonary cryptococcus infects the human body through lung invasion, but if it is not intervened in time, it may spread to other organs. The most serious case is the infection of the central nervous system which causes meningitis, and diffusive infection with cryptococcus is present in nearly all of the deaths (*Setianingrum, Rautemaa-Richardson & Denning, 2019*). Appropriate clinical intervention and treatment directly or indirectly affect the outcome of treatment, and PC is usually treated with antifungal drugs or corresponding symptomatic treatment (*Sloan & Parris, 2014*; *Williamson et al., 2017*). In our study, we analyzed the effect of other specific drugs on the prognosis of patients with PC, and found that the use of cytotoxic drugs was associated with an increased risk of poor prognosis in patients, while hormone therapy and immunosuppressive agents had no significant effect on the prognosis. These results suggest that PC patients with other complications or underlying diseases should carefully choose cytotoxic drugs in the context of a comprehensive evaluation of risks and benefits.

In the current study, 115 cases (21.9%) had a poor prognosis due to death or deterioration. Multivariate analysis showed that, among the prognostic factors, the decrease of albumin, platelet count, hematocrit, white blood cell count, PCT and CRP, as well as the increase of CD4/CD8 ratio, increased the risk of poor prognosis of PC. PCT and CRP are typical examination indicators of bacterial and pneumococcal infection, and are used in clinical diagnosis of pneumonia (*Dai et al., 2021*). The increase of these two indicators indicates that patients with pulmonary cryptococcal disease may also be complicated with other pulmonary disease infections, which will increase the risk of poor prognosis. CD4 and CD8 are related to the immune function of patients. Abnormalities in this index are common in HIV-positive or other immune diseases, and can significantly affect the prognosis in PC (*Lee et al., 2011*; *Majumder, Mandal & Bandyopadhyay, 2011*).

Although the risk factors that affect the prognosis of PC patients obtained in this study are similar to those reported in previous studies, this study differs from previous studies in terms of statistics and research methods. Most previous studies used single factor analysis and multivariable stepwise regression analysis to obtain influencing factors. However, in this process, various confounding factors may be considered as predictive variables due to multicollinearity and other problems. In contrast, this study adopted LASSO regression analysis to select the optimal variables, thus reducing the possible multicollinearity problem

in the model, and providing more accurate results than previous studies. Compared with the traditional logistic regression analysis, the nomogram is a simpler and more intuitive expression of the statistical analysis model, which makes the prediction model more concise and effective in quantifying risk and have higher value in clinical applications. The advantages of nomogram in clinical research and practical application have been widely verified (*Balachandran et al., 2015*; *Flanigan, Polcari & Hugen, 2011*; *Kattan, 2003*; *Touijer & Scardino, 2009*; *Yang, 2013*). At present, this study has established a predictive nomogram model for poor prognosis of PC patients for the first time, and verified the accuracy, stability and clinical applicability of the model through C-index, ROC curve and DCA curve in the training set and validation set.

Due to the lack of specificity in the clinical manifestations of PC, the diagnosis and treatment of PC itself are difficult. In particular, many complications caused by the poor prognosis, such as PC combined with cryptococcal meningitis, will greatly aggravate the patient's condition and risk degree. Therefore, more attention should be paid to the early diagnosis and standardized active treatment of PC, and the establishment of a personalized scoring system based on clinical symptoms and inspection indicators is conducive to guiding clinical diagnosis and prognosis evaluation. The predictive nomogram of poor prognosis of PC developed in this study has an ideal predictive effect. However, as this study is a single-center retrospective study with small sample size, and the source of the sample is limited, the conclusions still need to be further verified by a prospective multi-center cohort study with large sample. Additionally, the predictive variables came from clinical data, which may lead to research bias due to inconsistent and incomplete recording standards, and the results might have some limitations. Therefore, in the follow-up research work, it is also necessary to carry out cooperative research with other centers to improve the multi-center verification and optimization of the model.

## CONCLUSION

Neurological symptoms, hematocrit, white blood cell count, platelet count, CRP, PCT, albumin count, the ratio of CD4+ to CD8+ and cytotoxic drugs are independent predictors of poor prognosis of patients with PC. At the same time, the nomogram developed can accurately and reliably predict the incidence of poor prognosis of patients with PC. It has certain practical value and clinical guiding significance for the prognosis risk assessment of PC and is also conducive to improving the prognosis of patients with PC to a certain extent.

### Funding

This study was supported by a grant from the Key Discipline of Jiaxing Respiratory Medicine Construction Project (No. 2019-zc-04), the Jiaxing Key Laboratory of Precision Treatment for Lung Cancer, and the Key Construction Disciplines of Provincial and Municipal Co construction of Zhejiang (NO. 2023-SSGJ-002). The funders had no role in study design, data collection and analysis, decision to publish, or preparation of the manuscript.

## Grant Disclosures
The following grant information was disclosed by the authors:
Key Discipline of Jiaxing Respiratory Medicine Construction Project: 2019-zc-04.
Jiaxing Key Laboratory of Precision Treatment for Lung Cancer.
Key Construction Disciplines of Provincial and Municipal Co construction of Zhejiang: 2023-SSGJ-002.

## Competing Interests
The authors declare there are no competing interests.

## Author Contributions
- Xiaoli Tan conceived and designed the experiments, performed the experiments, analyzed the data, prepared figures and/or tables, authored or reviewed drafts of the article, and approved the final draft.
- Yingqing Zhang conceived and designed the experiments, performed the experiments, analyzed the data, prepared figures and/or tables, authored or reviewed drafts of the article, and approved the final draft.
- Jianying Zhou performed the experiments, authored or reviewed drafts of the article, and approved the final draft.
- Wenyu Chen conceived and designed the experiments, performed the experiments, analyzed the data, prepared figures and/or tables, authored or reviewed drafts of the article, and approved the final draft.
- Hua Zhou performed the experiments, authored or reviewed drafts of the article, and approved the final draft.

## Ethics
The following information was supplied relating to ethical approvals (*i.e.*, approving body and any reference numbers):

Ethical approval was granted by the Ethics Committee of the First Affiliated Hospital of Zhejiang University in China (ID: IIT20210761A).

## Data Availability
The raw data are available in the Supplementary Files.

## Supplemental Information
Supplemental information for this article can be found online at http://dx.doi.org/10.7717/peerj.17030#supplemental-information.

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
