# Peer review of "Construction and validation of a nomogram model to predict the poor prognosis in patients with pulmonary cryptococcosis"

_PeerJ, doi:10.7717/peerj.17030_

## Round 0.1 · original submission · Major Revisions

Dear authors,

After reading the reviews it was decided that you need to make a major revision to your manuscript. You should rewrite the manuscript in line with the reviewers' comments or write a detailed rebuttal on a point-by-point basis.

Reviewer 1 ·

Basic reporting

Np comment

Experimental design

No comment

Validity of the findings

Discussion should be rewritten. First start with the results you got and then compare each with results from previous studies. Also explain what is the underlying pathophysiological mechanism.
First sentence should be line 201 … - In this study….

Reviewer 2 ·

Basic reporting

1.Reference is needed for the sentence ‘The central nervous system infection caused by cryptococcus accounts for 80% of cryptococcal infection’ (Page 6, Line 53).
2. Spell out the word ‘PC’ at its first appearance. (Page 7, Line 54)
3. Please modify the expression more precisely as hematology malignance, not hematological tumors. Similarly, disseminated infection, not infection metastasis; neurological symptom, not psychiatric symptom.
4. The expression is quite confused as you said ‘the best treatment scheme is mainly to use antifungal drugs or select an appropriate therapy according to individual conditions in clinical practice’. What do you mean by appropriate therapy other than antifungal treatment?
5. There’re many effective methods to diagnose cryptococcosis including detection for cryptococcal antigen, lung biopsy, cerebral spinal fluid examination, et al. Despite non-specific clinical symptoms, I don’t think the diagnostic methods are limited as the author said. (Page 7, Line 64-65)

Experimental design

1. Cryptococcal meningitis is not included in the predictive model although it’s considered as a severe complication of pulmonary cryptococcosis. Why? And please clarify how the potential risk factors shown in Table 2 were identified, from statistical analysis or clinical experience only?
2. In method section, what do you mean by hormone therapy? Is it treatment with glucocorticoids?
3. The diagnostic criteria for cryptococcosis including pulmonary cryptococcosis and cryptococcal meningitis needs to be described as clearly as possible.
4. How long were the patents followed up?
5. How was the onset time defined in this study?

Validity of the findings

No comment

Additional comments

The author tried to establish a predictive model for poor prognosis of cryptococcosis. The major problems are shown above.

Reviewer 3 ·

Basic reporting

Dear Authors,
Thanks for your effort in this field. You constructed and validated a nomogram model to predict
the poor prognosis in patients with pulmonary cryptococcosis. The language was clear. The article included sufficient introduction and background to demonstrate how the work fits into the broader field of knowledge. The article structure was good.

Experimental design

The experimental design was appropriate. However, the study lacked an external validation cohort.

Validity of the findings

The impact and novelty of the article were general. You could add a table to detail the score based on the nomogram.

---

## Round 0.2 · Minor Revisions

Dear Authors,

Make revisions as suggested by reviewer #2 and return your revised paper or write a detailed rebuttal.

Reviewer 1 ·

Basic reporting

Everything was corrected as requested.

Experimental design

No comment.

Validity of the findings

Everything was corrected as requested.

Additional comments

Line 123 – ypical – please correct…

Reviewer 2 ·

Basic reporting

The manuscript is much improved.

Experimental design

Experimental design is modified as suggested.

Validity of the findings

Impact and novelty is fair.

Additional comments

1. Please correct the spelling error "ypical" (Page 5, Line 123).

Reviewer 3 ·

Basic reporting

The current edition of the revised manuscript could be accepted for publication.

Experimental design

There is no need to improve the design.

Validity of the findings

Those findings need to be validated in an external cohort. However, the author could not provide it now.

---

## Round 0.3 · accepted · Accept

Dear Authors, your manuscript is acceptable in its current form.